# Development of a Prediction Model for Positive Surgical Margin in Robot-Assisted Laparoscopic Radical Prostatectomy

Ying Hao [1,2], Qing Zhang [2], Junke Hang [2,3], Linfeng Xu [2], Shiwei Zhang [2] and Hongqian Guo [1,2,3,*]

1. Department of Urology, Nanjing Drum Tower Hospital Clinical College of Jiangsu University, Nanjing 210008, China
2. Institute of Urology, Nanjing University, Nanjing 210008, China
3. Department of Urology, Nanjing Drum Tower Hospital Clinical College of Nanjing Medical University, Nanjing 210008, China
* Correspondence: dr.ghq@nju.edu.cn; Tel.: +86-83106666

**Abstract:** A positive surgical margin (PSM) is reported to have some connection to the occurrence of biochemical recurrence and tumor metastasis in prostate cancer after the operation. There are no clinically usable models and the study is to predict the probability of PSM after robot-assisted laparoscopic radical prostatectomy (RALP) based on preoperative examinations. It is a retrospective cohort from a single center. The Lasso method was applied for variable screening; logistic regression was employed to establish the final model; the strengthened bootstrap method was adopted for model internal verification; the nomogram and web calculator were used to visualize the model. All the statistical analyses were based on the R-4.1.2. The main outcome was a pathologically confirmed PSM. There were 151 PSMs in the 903 patients, for an overall positive rate of 151/903 = 16.7%; 0.727 was the adjusted C statistic, and the Brier value was 0.126. Hence, we have developed and validated a predictive model for PSM after RALP for prostate cancer that can be used in clinical practice. In the meantime, we observed that the International Society of Urological Pathology (ISUP) score, Prostate Imaging Reporting and Data System (PI-RADS) score, and Prostate-Specific Antigen (PSA) were the independent risk factors for PSM.

**Keywords:** prediction model; positive surgical margin (PSM); robot-assisted laparoscopic radical prostatectomy (RALP)





## 1. Introduction

According to the 2020 global cancer statistics released by the International Agency for Research on Cancer (IARC), there were more than 1.4 million newly diagnosed cases of prostate cancer worldwide in 2020, accounting for 7.3% of new cancers, ranking third after breast cancer and lung cancer [1]. In China, the incidence of prostate cancer has steadily increased since 2015, owing to the continuous westernization of diet structure and lifestyle, as well as population aging [2–4]. However, compared with other types of cancer, prostate cancer is relatively inactive, and most patients have access to surgery with positive therapeutic effects. Consequently, the mortality rate of prostate cancer is much lower than the morbidity rate. Even in relatively advanced cases, neoadjuvant treatment does not preclude the possibility of achieving local benefits [5,6] to obtain the opportunity for surgery. Therefore, radical prostatectomy (whether laparoscopic or robot-assisted) remains the first-line treatment option for prostate cancer [7,8]. However, the effect of surgery varies greatly from one patient to the other, and there is still a phenomenon of biochemical recurrence or even tumor metastasis in 27–53% of patients after surgery, which, in extreme cases, can even be fatal to the patient [9]. Multiple studies have demonstrated that a positive surgical margin (PSM) has been in relationship to the phenomenon of biochemical recurrence and tumor metastasis in prostate cancer after the operation [10–12]. Some studies have also indicated that different surgical methods, surgical pathways, and resection levels

had an impact on the surgical margin. For example, robot-assisted laparoscopic radical prostatectomy (RALP) with Retzius preservation (or the posterior approach) has a more favorable prognosis regarding urinary incontinence but carries a greater risk of developing a PSM [13]. Nerve-sparing increases the risk of ipsilateral PSM [14]. The new anatomical tip dissection adopting the pubic prostatectomy collar opening technique may have a beneficial effect on the operative cutting edge if surgery is performed [15]. As a consequence of this, preoperative surgical margin judgment is a crucial part of the planning process for surgical procedures.

Currently, some studies on the related predictive factors of surgical margins of prostate cancer have been discovered. The following factors have been identified as independent predictors of PSM: preoperative PSA, biopsy Gleason score, percentage of positive biopsy needles, biopsy nerve infiltration, pathological Gleason score, pathological stage, lymph node positivity, the extracapsular extension of the tumor, and seminal vesicle infiltration [16–18]. Previously, some similar prediction models were established, but the models included a few factors or were only based on MR, or the score was relatively rough and not precise enough [19–21]. There has been no research done to help visualize the complicated model, which makes it inconvenient for both clinical and application work. As a result, we hope to obtain a more comprehensive and detailed model to compensate for the shortcomings of previous models. Moreover, we will illustrate the model using a nomogram. A web calculator will also be provided to make the model easier to use.

## 2. Materials and Methods

### 2.1. Study Design and Data Sources

We conducted a retrospective cohort study on a large population of patients with prostate cancer using patient data obtained from the Doctor's Work System of Nanjing Drum Tower Hospital. We included prostate cancer patients who underwent RALP in Nanjing Drum Tower Hospital from January 2018 to December 2021, and excluded patients who received preoperative neoadjuvant treatment, patients with a history of prostate-related surgery, patients who experienced prostate biopsy in an external hospital, and cases with important data missing. To rule out the possibility of metastasis, all patients underwent preoperative imaging examinations. Patients whose Gleason score ≥3+4 underwent lymphadenectomy. To make full use of the data, all of it was applied to establish the derived data set, and the enhanced bootstrap method was employed to conduct internal data verification.

### 2.2. Outcome

The outcome was a positive postoperative pathological margin. A positive margin is the extension of a cancer cell within the ink section of a RALP specimen [22]. The result of PSM will be determined after a staining procedure performed by an experienced pathologist. To ensure the authenticity of the data, the pathologist was not aware of the predictors' results when measuring the outcome.

### 2.3. Predictors

Based on previous research findings, we examined the factors that influence the surgical margin [16–18] and added some new variables. All predictors were measured by a qualified physician before surgery. To ensure the authenticity of the data, clinicians and test physicians were unaware of the outcomes when measuring the predictors, and each predictor was measured independently without mutual interference. Following are the details of the predictors.

Age: The patient's age at the time of surgery.

BMI: The patient's BMI at the time of hospitalization.

Prostate volume (V): The volume of the prostate measured at the time of the B-ultrasound-fused magnetic resonance prostate biopsy.

Percent of positive needles (PPN): The ratio of the total number of needles to the number of needles that reached tumor cells.

International Society of Urological Pathology (ISUP) score: According to the consensus of the classification meeting of the Society of Urological Pathology, the ISUP score was provided by prostate biopsy pathology [23].

Percent of Tumor (PT): The sum of the tumor fractions per needle in the tissue obtained through a prostate biopsy multiplied by 100.

Prostate Imaging Reporting and Data System (PI-RADS) score: It was provided by the suspicious nodule from the preoperative 3.0T MRI. The film was read and the results were given by a professional imaging doctor.

Tumor location (TL): The location of the suspicious nodule from the preoperative 3.0T MRI of the patients before surgery, which was read by a professional in medical imaging. It was grouped according to the peripheral zone (P), transitional zone (T), mixed (M), and Negative (prostate cancer is not currently being considered).

Maximal tumor diameter (D): The maximal diameter of the suspicious nodule that can be quantified in the preoperative 3.0T MRI of the patients.

The number of tumors (NT): The number of suspicious nodules on the preoperative 3.0T MRI of the patients.

Clinical staging of the tumor provided by MRI (T-MRI): It was evaluated on the basis of the preoperative 3.0T MRI and the pathological results of the prostate biopsy. It was categorized according to the latest eighth edition of tumor-staging criteria issued by the Joint Commission on Cancer (AJCC) [24]. Stages less than or equal to T2a are divided into group 1; T2b into group 2; the patients with staging greater than or equal to T2c were divided into group 3.

Prostate-Specific Antigen (PSA): PSA is valued before the patient's prostate biopsy (at the time of the initial diagnosis), with some missing data replaced by preoperative PSA values.

PSA density (PSAD): The portion of PSA in the prostate volume.

Inflammation index (II): Neutrophil count * platelet count/lymphocyte count. The information was obtained from the patient's preoperative routine blood test in order to evaluate preoperative inflammation.

Time from biopsy to surgery (t): The time between the prostate biopsy and the RALP. It was considered that the biopsy might cause local inflammation and edema of the prostate, which may affect the surgical margin.

Operator: The operator performing RALP for the patients. It was grouped according to the experience of RALP.

Others: Other risk factors, which predominantly include but are not limited to, nerve involvement by biopsy, patients with EPE or SVI, and patients with substantial clinical manifestations.

### 2.4. Sample Size

The sample size is determined by the amount of data available. Because the percentage of missing data was small (21/903 = 2.3%) and the type of missing data was significant, after eliminating the missing data, a total of 882 complete data were obtained.

### 2.5. Statistical Analysis Method

The potential predictors were determined through literature searches and discussions with experts at Nanjing Drum Tower Hospital. First, logistic regression was used for univariate analysis. After that, the variables were screened using the Lasso method. The Lasso method is a kind of compression estimation [25]. As the number of predictive variables in this study was more than the number of positive samples (according to the standard of EPV $\geq$ 10:1), collinearity was suspected among the predictive variables. It is necessary to use the Lasso method for screening variables to increase penalty terms, control collinearity, and avoid over-fitting to a certain extent. Based on the optimal lambda value,

nine prediction factors including age, PPN, ISUP, PT, D, PI-RADS, TL, T-MRI, and PSA were screened out for the establishment of the final logistic regression model. Since no treatment was performed on the patient, the interaction terms were not taken into account during the modeling process. Finally, the enhanced bootstrap method was used for internal verification. The nomogram was used to calculate the probability of each individual's prediction. To detail the assessment of the effect of model predictions, we will report model accuracy (C-statistic), model calibration (calibration map), and others (Brier score).

Since there was no standard probabilistic risk stratification as a point of reference, we divided the PSM risk of patients into four groups with low risk, low–medium risk, medium–high risk, and high risk by using the statistical concept and quartile.

## 3. Results

### 3.1. Study Population

In total, 903 patients met our inclusion criteria and 882 samples were used for modeling after excluding 21 with missing data. Figure 1 is a flowchart of sample inclusion and exclusion.

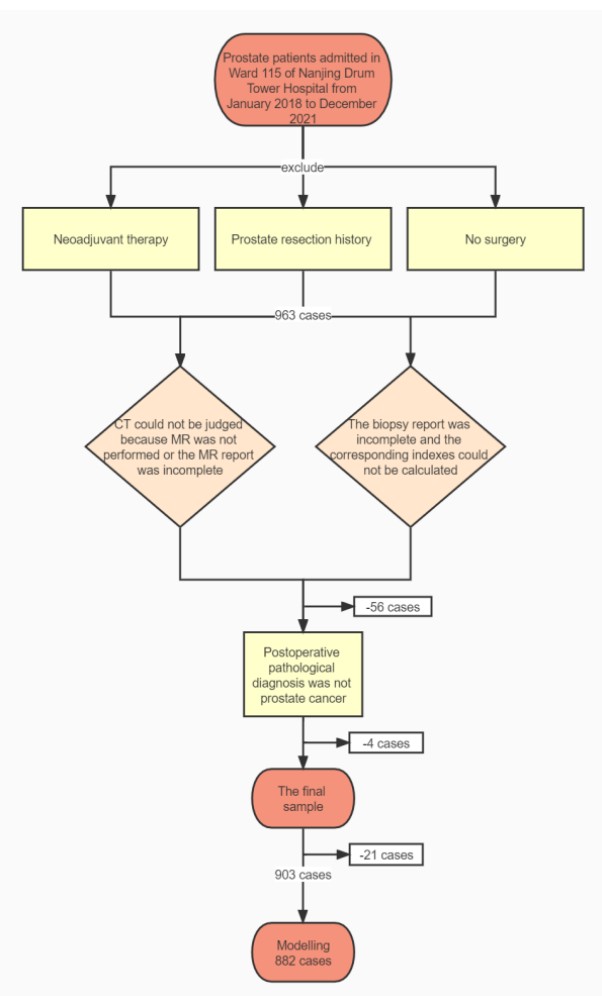

**Figure 1.** Inclusion and exclusion flowchart.

Table 1 indicates the baseline characteristics of overall and positive- and negative-margin patients. It is demonstrated that the age of the patients is concentrated in the range of 65 to 75 years, and as a whole, the majority of patients are in an early clinical stage, which is manifested as low histological score (ISUP) and MRI score (PI-RADS). At the same time, significant differences in age, PPN, ISUP, PT, PI-RADS, TL, D, T-MRI, PSA, PSAD, and Others can be observed between the margin-positive and margin-negative groups.

**Table 1.** Baseline characteristics of patients with prostate cancer. Values are numbers (percentages) of patients.

| | Level | Overall | NSM | PSM | *p* | Test | SMD |
|---|---|---|---|---|---|---|---|
| n | | 903 | 752 | 151 | | | |
| age (median [IQR]) | | 70.00 [66.00, 75.00] | 70.00 [65.00, 74.00] | 71.00 [66.00, 77.00] | 0.014 | nonnorm | 0.240 |
| BMI (mean (SD)) | | 24.60 (2.95) | 24.60 (2.93) | 24.58 (3.06) | 0.922 | | 0.009 |
| V (median [IQR]) | | 32.40 [24.70, 44.90] | 33.05 [24.78, 45.92] | 31.10 [24.20, 39.20] | 0.095 | nonnorm | 0.139 |
| PPN (median [IQR]) | | 0.36 [0.21, 0.50] | 0.34 [0.19, 0.50] | 0.44 [0.29, 0.67] | <0.001 | nonnorm | 0.559 |
| ISUP (%) | 1 | 235 (26.0) | 218 (29.0) | 17 (11.3) | <0.001 | | 0.587 |
| | 2 | 264 (29.2) | 230 (30.6) | 34 (22.5) | | | |
| | 3 | 230 (25.5) | 179 (23.8) | 51 (33.8) | | | |
| | 4 | 164 (18.2) | 118 (15.7) | 46 (30.5) | | | |
| | 5 | 10 (1.1) | 7 (0.9) | 3 (2.0) | | | |
| PT (median [IQR]) | | 14.29 [6.25, 25.94] | 12.50 [5.53, 23.47] | 24.29 [12.32, 37.71] | <0.001 | nonnorm | 0.669 |
| PI-RADS (%) | 3 | 193 (21.4) | 180 (23.9) | 13 (8.6) | <0.001 | | 0.673 |
| | 4 | 355 (39.3) | 311 (41.4) | 44 (29.1) | | | |
| | 5 | 318 (35.2) | 227 (30.2) | 91 (60.3) | | | |
| | N | 37 (4.1) | 34 (4.5) | 3 (2.0) | | | |
| TL (%) | M | 236 (26.1) | 185 (24.6) | 51 (33.8) | 0.056 | | 0.246 |
| | N | 32 (3.5) | 28 (3.7) | 4 (2.6) | | | |
| | *p* | 354 (39.2) | 307 (40.8) | 47 (31.1) | | | |
| | T | 281 (31.1) | 232 (30.9) | 49 (32.5) | | | |
| D (median [IQR]) | | 1.30 [0.90, 1.90] | 1.30 [0.90, 1.80] | 1.70 [1.20, 2.40] | <0.001 | nonnorm | 0.545 |
| NT (%) | 0 | 37 (4.1) | 33 (4.4) | 4 (2.6) | 0.520 | | 0.154 |
| | 1 | 546 (60.5) | 458 (60.9) | 88 (58.3) | | | |
| | 2 | 263 (29.1) | 217 (28.9) | 46 (30.5) | | | |
| | 3 | 51 (5.6) | 40 (5.3) | 11 (7.3) | | | |
| | 4 | 6 (0.7) | 4 (0.5) | 2 (1.3) | | | |
| T-MRI (%) | 1 | 483 (53.5) | 428 (56.9) | 55 (36.4) | <0.001 | exact | 0.426 |
| | 2 | 75 (8.3) | 55 (7.3) | 20 (13.2) | | | |
| | 3 | 345 (38.2) | 269 (35.8) | 76 (50.3) | | | |
| PSA (median [IQR]) | | 8.94 [6.40, 13.61] | 8.57 [6.11, 12.10] | 14.90 [8.02, 23.77] | <0.001 | nonnorm | 0.561 |
| PSAD (median [IQR]) | | 0.27 [0.18, 0.44] | 0.26 [0.17, 0.40] | 0.42 [0.27, 0.80] | <0.001 | nonnorm | 0.570 |
| II (median [IQR]) | | 376.68 [276.07, 519.53] | 382.54 [274.14, 515.20] | 366.61 [282.82, 541.58] | 0.965 | nonnorm | 0.022 |
| margin (%) | 0 | 752 (83.3) | 752 (100.0) | 0 (0.0) | <0.001 | | NaN |
| | 1 | 151 (16.7) | 0 (0.0) | 151 (100.0) | | | |
| t (median [IQR]) | | 14.00 [10.00, 18.00] | 14.00 [10.00, 18.00] | 13.00 [10.00, 18.00] | 0.571 | nonnorm | 0.019 |
| Operator (%) | 1 | 416 (46.1) | 339 (45.1) | 77 (51.0) | 0.397 | | 0.122 |
| | 2 | 409 (45.3) | 346 (46.0) | 63 (41.7) | | | |
| | 3 | 78 (8.6) | 67 (8.9) | 11 (7.3) | | | |
| Others (%) | 0 | 858 (95.0) | 722 (96.0) | 136 (90.1) | 0.004 | | 0.235 |
| | 1 | 45 (5.0) | 30 (4.0) | 15 (9.9) | | | |

V: Prostate volume; PPN: Percent of positive needles; ISUP, International Society of Urological Pathology; PT: Percent of Tumor; PI-RADS, Prostate Imaging Reporting and Data System; TL: Tumor location; D: Maximal tumor diameter; NT: Number of tumors; T-MRI: Clinical staging of the tumor provided by MRI. Stages less than or equal to T2a are divided into group 1; T2b into group 2; the patients with staging greater than or equal to T2c were divided into group 3; PSA: Prostate-Specific Antigen; II: Inflammation index; t: Time from biopsy to surgery; Others: Other risk factors.

For the new predictors worthy of concern, the median of PPN is 0.34 in the NSM group and 0.44 in the PSM group, which was distinguished by substantial differences. The same

difference appeared in the PT factor, with a median PT of 12.5 in the NSM group and 24.29 in the PSM group. Since tumors are often irregular in shape and their volumes are not easy to measure, we introduced the concept of the longest diameter. We were surprised to find a significant distribution, with a median of 1.3 for D in the NSM group and up to 1.7 in the PSM group. PSAD based on PSA performed similarly to PSA, possibly because the prostate volumes did not show distinctions between the two groups. However, the performance of NT, II, T, and Others was not satisfactory.

### 3.2. Modeling

There were only 151 events, and yet we considered 17 predictors. To satisfy the requirement that a factor must be assigned to at least 10 events, we began with univariate analysis. Table 2 shows the estimated values, standard errors, and *p*-values of the regression coefficients for univariate analysis. Finally, 10 factors including age, PPN, ISUP, PT, D, PI-RADS, PSA, PSAD, TL, and Others were selected for subsequent screening. Because clinical staging is a relatively important parameter for prostate cancer, we included it in subsequent screening, although it was not significant in univariate analysis.

**Table 2.** Regression coefficients for univariate analysis.

| Coefficients: | Univariate Analysis | | |
|---|---|---|---|
| | **Estimate** | **Std. Error** | **Pr (> | z | )** |
| age | 0.039 | 0.014 | 0.006 *** |
| BMI | −0.019 | 0.031 | 0.546 |
| V | −0.007 | 0.005 | 0.159 |
| PPN | 2.266 | 0.4.5 | $2.13 \times 10^{-8}$ *** |
| ISUP-2 | 0.573 | 0.315 | 0.069 * |
| ISUP-3 | 1.270 | 0.299 | $2.13 \times 10^{-5}$ *** |
| ISUP-4 | 1.480 | 0.311 | $1.90 \times 10^{-6}$ *** |
| ISUP-5 | 1.681 | 0.735 | 0.022 ** |
| PT | 0.040 | 0.006 | $1.8 \times 10^{-11}$ *** |
| D | 0.693 | 0.115 | $1.67 \times 10^{-9}$ *** |
| PI-RADS-4 | 0.551 | 0.322 | 0.087 * |
| PI-RADS-5 | 1.544 | 0.306 | $4.5 \times 10^{-7}$ *** |
| PI-RADS-N | −0.313 | 0.779 | 0.688 |
| NT-1 | 0.417 | 0.543 | 0.443 |
| NT-2 | 0.515 | 0.555 | 0.354 |
| NT-3 | 0.819 | 0.630 | 0.193 |
| NT-4 | 1.705 | 1.055 | 0.106 |
| PSA | 0.026 | 0.009 | 0.006 *** |
| PSAD | 1.263 | 0.220 | $9.59 \times 10^{-9}$ *** |
| T-MRI-2 | 0.070 | 0.373 | 0.852 |
| T-MRI-3 | 0.235 | 0.246 | 0.342 |
| TL-N | −1.484 | 0.745 | 0.046 ** |
| TL-*p* | −0.607 | 0.231 | 0.009 *** |
| TL-T | −0.127 | 0.224 | 0.574 |
| II | 0.0002 | 0.000 | 0.692 |
| t | −0.001 | 0.004 | 0.796 |
| Others-1 | 0.860 | 0.345 | 0.013 ** |
| Operator 100–200 cases | −0.220 | 0.191 | 0.248 |
| Operator <100 cases | −0.387 | 0.363 | 0.286 |

* $p < 0.1$; ** $p < 0.05$; *** $p < 0.01$, Univariate analysis was based on the logistic model.

After screening by the Lasso method, we obtained the final prediction factors and the final prediction model. Table 3 demonstrates the regression coefficient estimates, OR, 95% CI, and *p*-values for the final prediction model. Nomograms, as depicted in Figure 2, will be used to visualize models for the convenience of clinicians.

**Table 3.** The prediction model.

| Coefficients | Estimate | OR | 95% Confidence Interval | | Pr (> \|z\|) |
|---|---|---|---|---|---|
| age | 0.020 | 1.020 | 0.991 | 1.051 | 0.180 |
| PPN | 0.696 | 2.006 | 0.424 | 9.500 | 0.381 |
| ISUP-2 | 0.260 | 1.297 | 0.673 | 2.502 | 0.438 |
| ISUP-3 | 0.630 | 1.877 | 0.968 | 3.636 | 0.063 * |
| ISUP-4 | 0.846 | 2.329 | 1.172 | 4.630 | 0.016 ** |
| ISUP-5 | 1.243 | 3.466 | 0.742 | 16.202 | 0.115 |
| PT | 0.017 | 1.017 | 0.993 | 1.043 | 0.169 |
| D | 0.076 | 1.079 | 0.747 | 1.559 | 0.687 |
| PI-RADS-4 | 0.349 | 1.417 | 0.727 | 2.762 | 0.306 |
| PI-RADS-5 | 0.699 | 2.012 | 0.979 | 4.135 | 0.058 * |
| PI-RADS-N | −1.358 | 0.257 | 0.013 | 4.907 | 0.367 |
| PSA | 0.026 | 1.026 | 1.008 | 1.045 | 0.006 *** |
| T-MRI2 | 0.070 | 1.072 | 0.516 | 2.228 | 0.852 |
| T-MRI3 | 0.235 | 1.264 | 0.780 | 2.050 | 0.342 |
| TL-N | 1.357 | 3.883 | 0.211 | 71.321 | 0.361 |
| TL-$p$ | −0.360 | 0.697 | 0.408 | 1.193 | 0.189 |
| TL-T | 0.430 | 1.537 | 0.908 | 2.604 | 0.110 |
| Intercept | −5.203 | 0.005 | | | 0.00001 *** |

Observations: 882.000, Log Likelihood: −337.820, Akaike Inf. Crit: 711.639, * $p < 0.1$; ** $p < 0.05$; *** $p < 0.01$, OR and 95% CI were calculated by SPSS 22.

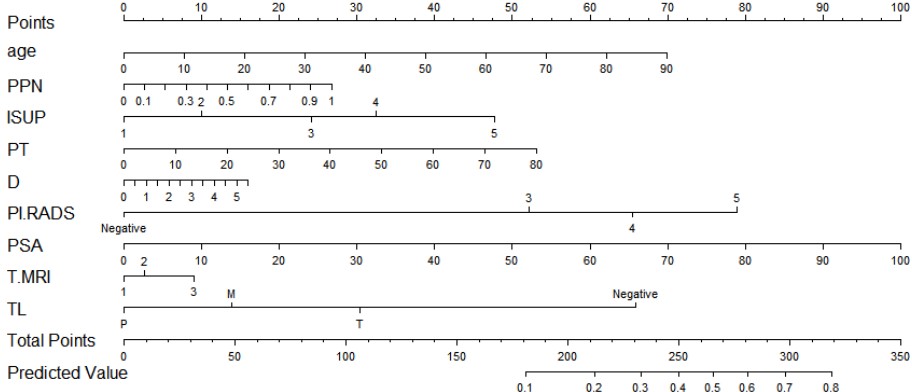

**Figure 2.** The nomogram.

To make it easier for using, we also provide a web calculator. This is the website for the calculator. https://doctor-h.shinyapps.io/dynnomapp/ (accessed on 23 November 2022).

As shown in Figure 2, If a patient was 70 years old, the ratio of positive needles was 0.5, the histological grade was 3, and the total tumor ratio was 30. The longest diameter of the suspicious nodule was 1.53, the suspicious nodule score was 4, the PSA was 7.55, the location of the suspicious nodule was a transitional zone, and the clinical stages were 2c and above. With a total score of 385 and a PSM probability of 0.276, they belonged to the low-risk group with PSM.

The discrimination of the final model was calculated by the C statistic (this was the model discrimination index, and the greater the value, the better the discrimination). The C statistic upon the model establishment was 0.764 (0.722, 0.806), and the C statistic after internal verification and adjustment was 0.727. Moreover, the degree of calibration is shown as the Brier value (a measure of the model's degree of calibration, with a lower value indicating a better degree of calibration, generally less than 0.25), which is 0.118 after modeling and 0.126 after adjustments by internal validation. The calibration curve is shown in Figure 3. It could be observed that the model had a suitable replacement in the low-risk and medium-risk groups, and a generally acceptable fit in the medium-risk and high-risk groups, which would overestimate the result to some extent.

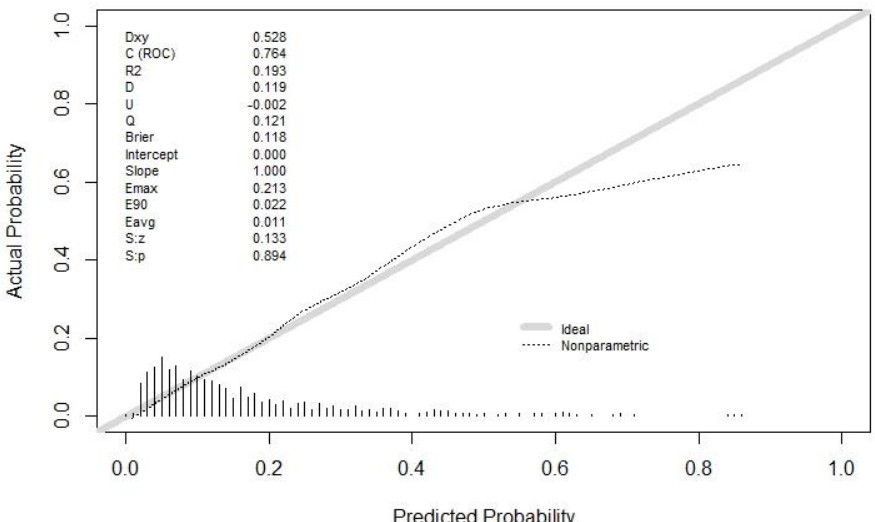

**Figure 3.** The calibration curve. Both the C statistic and Brier value in the figure are precalibration values. The C statistic after internal verification and adjustment was 0.727. The Brier value was 0.126 after internal validation adjustments. Both of them are acceptable.

Pathology information is provided as Appendix A (Table A1).

## 4. Discussion

We have developed and validated models to predict the risk of PSMs after RALP in patients with prostate cancer. This algorithm combines the influencing factors mentioned in previous studies with new variables associated with an increased risk of PSM. These included age, PPN, ISUP, PT, D, PI-RADS, TL, T-MRI, and PSA. The final model has been generated (Table 3). The model performed comparably in the validation and development data sets, and we deemed it clinically available.

The widely accepted view that prostate cancer is an age-related disease has been confirmed by numerous studies. As early as 2010, the American Cancer Society suggested that men with an average risk should receive information about the uncertainty, risk, and potential benefits of prostate cancer screening from age 50, while men in high-risk groups should receive this information before age 50. Based on the information, the patient decides whether to accept the examination [26]. Age has also been found to be an independent predictor of shorter prostate cancer-specific survival in men diagnosed with metastatic prostate cancer, even in an era of more effective treatment [27]. Now, we have also discovered and reported that age is a predictor of PSM after RALP in patients with prostate cancer. Some researchers have pointed out a male-specific association between the accumulation of DNA damage related to mutation and aging and the prostate cancer biomarker poly (ADP-ribose)-polymerase (PARP) [28]. This may be one of the reasons and corresponding research can be conducted to investigate it.

ISUP is the grading standard of prostate cancer for judging the malignancy and the prognosis of tumors established by the International Society of Urological Pathology, which is generally accepted to be the case that, the larger the subgroup, the worse the prognosis will be [29]. Such trends are also present in our model, with higher subgroups suggesting a higher risk of PSMs.

PI-RADS is an overall score of suspected prostate cancer nodules based on multi-parameter magnetic resonance imaging (mpMRI) proposed by the American Radiological Society (ACR) and the European Society of Urological Surgery (ESUR). The second edition published in 2016 is the latest standard for mpMRI imaging and reporting [30]. However, due to distinct equipment and the reporter's experience, there may still be bias. Overall, the PI-RADS score was consistent with the direction of risk for PSM. Upon examining the

regression model, we discovered that a PI-RADS score of 5 had a significant impact on the outcomes.

Tumor location was grouped according to previous studies [18]. The location of the tumor within the transitional zone was associated with an increased risk of PSM.

Compared with other studies using postoperative pathological staging [16], we prefer to use clinical staging (T-MRI) because it is available preoperatively. The criteria for T-MRI were by the AJCC cancer-staging 8th edition [24]. As a result, a higher T-MRI implies a higher risk of PSM. This may also explain the relevance of neoadjuvant therapy in patients with locally advanced stages, as the tumor may shrink to reduce T-MRI after neoadjuvant therapy. Surgical treatment at this time can reduce the risk of PSM.

PSA, as an essential biomarker of prostate cancer, has been widely used for the screening of various prostate cancers. Hereby, we reported that the probability of a high PSA level is relatively high in patients with a high risk of PSM.

In addition, the new indicators contain an intriguing element. The proportion of positive biopsy needles (PPN) can be approximated as a random sample, so a higher PPN means a larger tumor with a higher likelihood of cutting the tumor. This is in line with our observation that the risk of PSM rises as PPN levels rise. This phenomenon is also similar to the results reported in other studies [17]. Percentage of Tumor (PT) is a new concept we proposed based on PPN. We present this concept, because there may be a large number of positive needles in a biopsy but a small amount of tumor per needle. We expected it to perform better than PPN, but the result was not ideal. Although $p$ (0.169) for PT was less than PPN (0.381), indicating that PT was more likely to influence the results of PSM than PPN, its regression coefficient (0.017) was less than PPN (0.696), indicating that PT contributed less to the results than PPN. A larger sample size may then be required to verify the relevance of the PT. The tumor maximum diameter (D) is a new measurement we brought forward. The majority of the tumors are irregular and their volume is difficult to measure, which is the primary motivation for proposing this index. To date, no investigator has included it as a predictor of PSM. In the subgroup comparison, we obtained that D's distribution in the PSM group and the NSM group was significantly different, but after being included in the regression model, D's contribution was not optimistic. However, we can still find the influence of the change of D on the risk of PSM. This may be an approximation of the tumor's volume. Similarly, a larger sample size may subsequently be required to verify the importance of D.

The predictive factors included in this study are all indicators in the preoperative necessary examinations for clinical patients, which were easily acquired and explained in clinical work. For junior doctors who still lack clinical experience, this model can be regarded as a tool for their rapid adaptation to clinical work. Meanwhile, the inclusion of more predictors makes PSM more specific and predictable. If it extends from prostate cancer to other systems, it is reasonable to believe that similar research can be conducted to improve the therapeutic value of surgery for cancer in general.

The methods and standards for the derivation and validation of prediction models are presented in the 2015 TRIPOD Statement by the BMJ [31]. Its advantage is to make the research logic and process more rigorous, to make the report content more comprehensive and standardized, and to make different prediction models comparable. Currently, no study has pointed out any glaring flaws in the TRIPOD statement. The key advantages of the prediction model established in this study include: it covered the entire contents of the preoperative examination of prostate cancer, and these variables are easy to measure. The greatest advantage lies in the visualization of the model, which transforms abstract variables into practical evaluation tools. The limitations of this study include the possibility of bias due to insufficient valid data and information bias: First, because the rate of PSM is comparatively low (16.7%), which implies that inadequate relative sample size may cause some deviations from the results. In this instance, univariate analysis was considered to pre-screen factors. Although this approach has been questioned by statisticians, there seems to be no better solution to solve similar dilemmas. Second, most patients included in this study

are in the early clinical stage. This may allow the study model to perform effectively in early-stage patients and may perform poorly in the overall patient population. Eventually, additional validation studies may be required to cover a larger patient population.

Modeling and validation are currently performed on the same set of practices and individuals, while an independent validation study would be a more rigorous test that should be performed. We consequently expect other researchers interested in this research to apply data from different institutions and even divergent races to verify this model.

## 5. Conclusions

In conclusion, we have developed and validated a predictive model for PSM after RALP for prostate cancer that can be used in clinical practice. Simultaneously, we found that the ISUP score, PI-RADS score, and PSA were the independent risk factors for PSM. In this model, previous studies' factors were referred to, and some new factors were proposed to predict more comprehensively. This is probably the most relatively comprehensive and operable prediction model in this field. Although it still has certain limitations, the discrimination and calibration performance of the model are acceptable overall.

**Author Contributions:** Y.H.: methodology, software, validation, formal analysis, investigation, data curation, writing—original draft, visualization. Q.Z.: conceptualization, Resources, writing—review and editing. J.H.: investigation, data curation. L.X.: conceptualization, resources. S.Z.: conceptualization. H.G.: resources, supervision, project administration, funding acquisition. All authors have read and agreed to the published version of the manuscript.

**Funding:** This research was funded by the Chinesisch-Deutsches Mobilitätsprogramm, grant number M-0670.

**Institutional Review Board Statement:** Not applicable.

**Informed Consent Statement:** Informed consent was obtained from all subjects involved in the study.

**Data Availability Statement:** Not applicable.

**Conflicts of Interest:** The authors declare no conflict of interest. The funders had no role in the design of the study; in the collection, analyses, or interpretation of data; in the writing of the manuscript; or in the decision to publish the results.

## Appendix A

**Table A1.** Pathology Information.

|  |  | NSM | PSM | *p* |
|---|---|---|---|---|
| n |  | 752 | 151 |  |
| ML (%) | Apex | 0 | 45 (33.1) | 0.143 |
|  | Periphery | 0 | 42 (30.9) |  |
|  | Base | 0 | 23 (16.9) |  |
|  | Apex + Periphery | 0 | 12 (8.8) |  |
|  | Base + Periphery | 0 | 6 (4.4) |  |
|  | Base + Apex | 0 | 4 (2.9) |  |
|  | Apex + Base + Periphery | 0 | 3 (2.2) |  |
|  | Periphery + Spermaduct | 0 | 1 (0.7) |  |
| Gleason (%) | 3 + 3 | 105 (14.0) | 3 (2.0) | <0.001 |
|  | 3 + 4 | 394 (52.4) | 71 (47.0) |  |
|  | 3 + 5 | 1 (0.1) | 0 (0.0) |  |
|  | 4 + 3 | 194 (25.8) | 51 (33.8) |  |
|  | 4 + 4 | 41 (5.5) | 16 (10.6) |  |
|  | 4 + 5 | 14 (1.9) | 9 (6.0) |  |
|  | 5 + 3 | 2 (0.3) | 1 (0.7) |  |
|  | 5 + 4 | 1 (0.1) | 0 (0.0) |  |

**Table A1.** *Cont.*

| | | NSM | PSM | *p* |
|---|---|---|---|---|
| N (%) | 1 | 396 (52.7) | 100 (66.2) | 0.1 |
| | 2 | 220 (29.3) | 28 (18.5) | |
| | 3 | 61 (8.1) | 10 (6.6) | |
| | m | 75 (9.9) | 13 (8.6) | |
| TL (%) | M | 160 (21.3) | 47 (31.1) | <0.001 |
| | *p* | 380 (50.5) | 40 (26.5) | |
| | T | 212 (28.2) | 64 (42.4) | |
| EPE (%) | 0 | 504 (67.0) | 45 (29.8) | <0.001 |
| | 1 | 248 (33.0) | 106 (70.2) | |
| SVI (%) | 0 | 720 (95.7) | 123 (81.5) | <0.001 |
| | 1 | 32 (4.3) | 28 (18.5) | |
| VI (%) | 0 | 730 (97.1) | 139 (92.1) | 0.006 |
| | 1 | 22 (2.9) | 12 (7.9) | |
| NI(%) | 0 | 332 (44.1) | 25 (16.6) | <0.001 |
| | 1 | 420 (55.9) | 126 (83.4) | |
| Lymph node (%) | 0 | 288 (38.3) | 86 (57.0) | |
| | 1 | 11 (1.5) | 9 (6.0) | |
| | 2 | 453 (60.2) | 56 (37.1) | |
| pT (%) | T2 | 503 (66.9) | 43 (28.5) | <0.001 |
| | T3a | 217 (28.9) | 79 (52.3) | |
| | T3b | 32 (4.3) | 27 (17.9) | |
| | T4 | 0 (0.0) | 2 (1.3) | |

ML: Margin location; N: Number of the tumor; TL: Tumor location; VI: Vascular invasion; NI: Invasion of nerve; SVI: Seminal vesicle invasion.

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
