# Peer review of "Development of a Prediction Model for Positive Surgical Margin in Robot-Assisted Laparoscopic Radical Prostatectomy"

_curroncol, doi:10.3390/curroncol29120751_

Round 1
Reviewer 1 Report
I don’t believe this manuscript should be considered for publication. It is difficult to be followed and in consequence, to be used in clinical practice. Some considerations:
-Substantial English revision is required
-Please be careful with the use of abbreviations, that need to be explained. Moreover, the abbreviation RALRP in not used generally (please use RALP)
-How were patients diagnosed? Please provide more information about biopsy scheme. Were all targeted biopsies?
-Was MRI performed before biopsy or after?
-I believe the model is overfitted, difficult to be interpreted and some variables are redundant (PSAD with all prostate measures)
-Calibration of the model is suboptimal
-Results are difficult to be followed
-More information about margins are needed (size, location ISUP etc)
Author Response
Dear Reviewer:
Thank you very much for your comments concerning our manuscript entitled “Development of prediction model for positive surgical margin in robot-assisted laparoscopic radical prostatectomy”. Those comments are all valuable and very helpful for revising and improving our paper. We have studied the comments carefully and have made corrections which we hope meet with approval. The responses are as following.
- Comment: How were patients diagnosed? Please provide more information about the biopsy scheme. Were all targeted biopsies?
Response: The patient was diagnosed by biopsy based on transrectal ultrasound combined with MRI. The biopsy scheme was a 12-needle systematic puncture plus a 2-4 needle targeted puncture. About 15% of the patients, all of whom had Gleason scores of 6, underwent systemic puncture only.
- Comment: Was MRI performed before biopsy or after?
Response: MRI was performed before biopsy, and the target puncture site was determined according to the suspicious nodules suggested by MRI.
- Comment: I believe the model is overfitted, difficult to be interpreted and some variables are redundant (PSAD with all prostate measures)
Response: The fit of the model can be explained by the fact that the variables ultimately included in the model have their aspects that suggest the results. PSAD was a proposed variable based on PSA and prostate volume, intended to eliminate the effect of volume (because a larger prostate might mean more PSA secretion), and was not included in the model because it was screened out.
- Comment: Calibration of the model is suboptimal.
Response: The calibration degree of the model is reflected by the Brier value. The lower the value is, the better the calibration degree is. Generally, it is acceptable if the value is lower than 0.25. The Brier value of the model we built is 0.118 and 0.126 after adjustment. The calibration may not be optimal, but it is acceptable.
- Comment: Results are difficult to be followed.
Response: We have changed the expression of the results.
- Comment: More information about margins are needed (size, location ISUP, etc).
Response: Because the result of this study is: whether the margin is positive or not other information about the margin was not considered. According to your suggestion, we have submitted relevant information as supplementary material.
We tried our best to improve the manuscript and made some changes in the manuscript. These changes will not influence the content and framework of the paper. We appreciate your warm work earnestly and hope that the responses will meet with approval.
Once again, thank you very much for your comments and suggestions.
Yours sincerely,
Ying Hao
Reviewer 2 Report
The authors employed the Lasso method to identify the predictive factors for the final logistic regression model and a nomogram for visualization to predict the positive surgical margin in robot-assisted laparoscopic radical prostatectomy. However, there still are concerns that need to be addressed.
1. Author included multiple clinic characteristics into the regression model to predict positive surgical margin. The Lasso method was used to perform a screening to screen the clinic characteristics. It is unclear why author chose Lasso model for univariate analysis rather than the logistic model. The results of these two models differ?
2. The author noted that new variables such as PPN, PT and D, were added into the predict model. It may be ambitious. Based on the table 3 of multivariate analysis, only the grade and PSA are significant in the predict model. The rest factors are not significant in the model, indicating these factors are not independent predictors, may not significantly contribute to predict model.
3. The authors mentioned that different surgical methods, surgical pathways, and resection levels had an impact on the surgical margin. is there a specific characteristic for robot-assisted laparoscopic radical prostatectomy compared to other methods to predict PSM?
4. Is there a significant difference of ROC between authors’ model and other predict models?
Author Response
Dear Reviewer:
Thank you very much for your comments concerning our manuscript entitled “Development of prediction model for positive surgical margin in robot-assisted laparoscopic radical prostatectomy”. Those comments are all valuable and very helpful for revising and improving our paper. We have studied the comments carefully and have made corrections which we hope meet with approval. The responses are as following.
- Comment: Why author chose Lasso model for univariate analysis rather than the logistic model?
Response: Lasso method is a kind of compression estimation. Because the number of predictive variables in this study was more than the number of positive samples (according to the standard of EPV≥10:1), it was considered that the predictive variables might have collinearity. It is necessary to use the lasso to increase penalty terms, control collinearity, and screen variables, and at the same time avoid over-fitting to a certain extent.
- Comment: Some factors are not significant in the model, indicating these factors are not independent predictors, may not significantly contribute to predict model.
Response: The variables in the prediction model were screened by the Lasso method. In the lasso process, we choose the minimum value of λ, that is, the optimal solution. This indicates that the optimal model can be obtained by including these variables. We also used logistic regression to screen variables, taking the minimum AIC (meaning the best predictive performance) and obtained similar but not identical results. However, considering that our data are high-dimensional (the number of independent variables is larger than the number of samples), we finally adopted the variables selected by the Lasso method for the model establishment, even though some factors are not significant in the traditional sense.
- Comment: Is there a specific characteristic for robot-assisted laparoscopic radical prostatectomy compared to other methods to predict PSM?
Response: At present, studies have only shown that surgical methods, approaches, and excision levels are correlated with surgical margins, and no researchers have used them to predict surgical margins. The purpose of our paper mentioning this content is to remind clinicians that when preoperative examination indicates that patients have a high probability of positive margin, various surgical methods should be considered to avoid PSM to some extent. Surely, robot-assisted laparoscopic radical prostatectomy can improve surgical accuracy to a greater extent due to the use of robots. And the 3D three-dimensional makes the structure more clear. Perhaps in the future, some parameters in the surgery can be used to predict the positive margin. However, we believe that the clinical significance may not be significant, as pathology reports will be available immediately after the surgery to determine the margin. At this time, there is no good way to intervene. Patients can only be followed up for observation and continue with other treatment options if there is a biochemical recurrence. We expect the surgery to achieve its maximum therapeutic effect, which is why we chose the preoperative examinations for prediction.
- Comment: Is there a significant difference in ROC between authors’ model and other predict models?
Response: At present, few researchers have made prediction models of positive margin, and most of them only analyze the predictive factors and independent factors of PSM or analyze the influence of one factor on the positive margin separately. There is not much difference in ROC between ours and the other predictive models. But some of their models are complex (with separate MRI images), or just include variables and do not filter them. In short, they are not convenient for clinical use. The biggest difference in our study is the visualization of the model, which makes the complicated formula simple and clear for clinical use. In this revision, we have added a web calculator, which makes the use of the model much easier. And we strictly follow the TRIPOD guidelines (more standardized).
We tried our best to improve the manuscript and made some changes in the manuscript. These changes will not influence the content and framework of the paper. We appreciate your warm work earnestly and hope that the responses will meet with approval.
Once again, thank you very much for your comments and suggestions.
Yours sincerely,
Ying Hao
Reviewer 3 Report
The authors developed a model for the prediction of positive surgical margins after laparoscopic robot-assisted radical prostatectomy.
The paper is hard to follow and many sentences are confusing. The results are not clearly presented. The advantage of the paper includes the sample size and the modality of surgery (RALP). The topic is also clinically relevant. The study design is fine, but the quality of results’ presentation is poor. The aim of the study was to create a nomogram, but the presentation of the nomogram in the paper is poor. The nomogram should include only significant variables, independently associated with PSM. The presented nomogram includes many unnecessary variables and graphical presentation is blurred.
1. Abstract requires very profound changes. The results do not mention the factors related to PSM and this should be the most important message of the paper. Many language errors require correction.
2. In the materials and methods section the performance of imaging studies before surgery to exclude metastasis and most importantly the performance of lymphadenectomy should be mentioned.
3. Abbreviations should be explained when they are used for the first time in text and abstract.
Abbreviations should not be overused as it is done in the results and table 1.
Some abbreviations are not intuitive and must be changed (e.g. “D” is used for maximal tumour diameter). “N” is commonly used for lymph node staging.
4. In the table 1 cT staging is mostly 1 or 3. How to explain that only 8.3% of pts have cT2 disease and 38.2 cT3. I found the information that this is based on MRI and biopsy in the appendix. This coding of cT1-3 is very confusing and clinical staging is still based on DRE. If you report MRI findings you should use different abbreviation. Moreover according to your cT classification ,none of the patients has EPE or SVI. It is highly inexplicable. EPE and SVI would be suspected to be strong predicitiors of PSM but you do not include them.
5. What are regression coefficient estimates in table 3? Why logistic regression is used only for multivariable analysis and not for univariate?
If logistic regression is used, odds ratios with 95% confidence intervals should be presented in the table 3 instead of coefficient and standard error.
The multivariable analysis should include only the variables which are statistically significant.
Is only PSA, ISUP and PIRADS 5 on MRI independently associated with PSM?
If yes why nomogram includes all other variables? Nomogram should include only significant variables.
6. Figure and tables’ descriptions are too short and not informative enough.
7. The nomogram figure includes factors that appeared to be not relevant (P>0.05) in the multivariable analysis. It should include only relevant factors. The nomogram figure is blurred and includes a lot of unnecessary information. The idea of creating nomogram is very good, but this graphical presentation is poor and the nomogram is hard to use.
8. The authors use the term ‘prostatic extravasaction’. Did You mean extracapsular extension of the tumour?
9. Another table with postoperative variables (final histopathology) should be presented.
Author Response
Dear Reviewer:
Thank you very much for your comments concerning our manuscript entitled “Development of prediction model for positive surgical margin in robot-assisted laparoscopic radical prostatectomy”. Those comments are all valuable and very helpful for revising and improving our paper. We have studied the comments carefully and have made corrections which we hope meet with approval. The responses are as following.
- Comment: Abstract requires very profound changes.
Response: The summary has been revised as suggested.
- Comment: In the materials and methods section the performance of imaging studies before surgery to exclude metastasis and most importantly the performance of lymphadenectomy should be mentioned.
Response: The materials and methods have been revised as suggested.
- Comment: Abbreviations should be explained when they are used for the first time in text and abstract.
Response: The abbreviations have been revised as suggested.
- Comment: How to explain that only 8.3% of pts have cT2 disease and 38.2 cT3.
Response: Clinical staging was mainly based on MRI (given the new abbreviation cT-MRI) and was regrouped with group 1 ≤T2a, group 2 =T2b, and groups 3 ≥T2c. Because group 2 only included stage T2b, the number of cases was small. Because of the popularity of screening at Nanjing Drum Tower Hospital, most patients are in the early or middle stages. This results in little extracapsular invasion or seminal vesicle invasion in patients (based on preoperative examination). In some patients, nerve invasion was detected by biopsy. They were uniformly defined as Others, indicating other risk factors.
- Comment: What are regression coefficient estimates in table 3? Why logistic regression is used only for multivariable analysis and not for univariate? Is only PSA, ISUP, and PIRADS 5 on MRI independently associated with PSM? If yes why nomogram includes all other variables? Nomogram should include only significant variables.
Response: The regression coefficient estimate is just the regression coefficient, and it's called the coefficient estimate because it is calculated by the sample. Univariate analysis was also performed by logistic regression, and the lasso method was only used for further variable screening after univariate analysis. Logistic regression was not used for variable screening because the number of predictive variables in this study was more than the number of positive samples (according to the standard of EPV≥10:1), considering the possibility of collinearity of predictive variables. It is necessary to use the lasso method to increase penalty terms, control collinearity, and avoid over-fitting to a certain extent.
We selected variables with significance in univariate analysis. We included cT-MRI because we believed that it might affect the incisor margin, after comprehensive consideration and several discussions, although it was not significant in univariate analysis. Subsequent lasso screening did leave cT-MRI.
PSA, IUSP, and PIRADS were independently correlated with PSM because they were significant in multifactor logistic regression.
The nomogram is a visualization of the final predictive model and should contain all the variables in the final model. The variables in the prediction model were screened by the Lasso method. In the lasso process, we choose the minimum value of λ, that is, the optimal solution. This indicates that the optimal model can be obtained by including these variables. We also used logistic regression to screen variables, taking the minimum AIC (meaning the best predictive performance) and obtained similar but not identical results. However, considering that our data are high-dimensional (the number of independent variables is larger than the number of samples), we finally adopted the variables selected by the Lasso method for the model establishment, even though some factors are not significant in the traditional sense.
- Comment: Figure and tables’ descriptions are too short and not informative enough.
Response: Figures and tables’ descriptions have been added as suggested.
- Comment: The nomogram figure is blurred and includes a lot of unnecessary information. The idea of creating nomogram is very good, but this graphical presentation is poor and the nomogram is hard to use.
Response: According to the suggestion, we have simplified the information of the nomogram and attached a web calculator to make it easier to use.
- Comment: The authors use the term ‘prostatic extravasaction’. Did You mean extracapsular extension of the tumour?
Response: Yes. ‘Prostatic extravasaction’ means extracapsular extension of the tumour. Researchers are suggesting that prostatic extravasation is one of the predictors of positive margin. We quoted it directly from the article.
- Comment: Another table with postoperative variables (final histopathology) should be presented.
Response: According to the suggestion, post-operative pathology will be provided in supplementary materials.
We tried our best to improve the manuscript and made some changes in the manuscript. These changes will not influence the content and framework of the paper. We appreciate your warm work earnestly and hope that the responses will meet with approval.
Once again, thank you very much for your comments and suggestions.
Yours sincerely,
Ying Hao
Round 2
Reviewer 1 Report
Comments addressed
Author Response
Dear reviewer:
We would like to express our great appreciation again for your constructive comments on our manuscript.
Thank you very much.
Best wishes.
Ying Hao
Reviewer 2 Report
can be accepted in present form
Author Response

(The authors gave the same response as above.)

Reviewer 3 Report
Thank You for your answers.
Firstly, I am still not satisfied with the inclusion of so many statistically insignificant variables in the nomogram. I agree that sometimes clinically significant variables might be included in the nomogram despite being not statistically significant. In this case I do not believe that so many variables derived from MRI which are of not well-established value should be used in the nomogram (e.g. D, NT). Moreover using so many variables in the nomogram makes it less clinically useful.
Secondly, I still do not understand why the majority of patients is cT1 stage on MRI imaging - does it mean that no PIRADS lesions where identified? If yes, why PIRADS >= is provided for >90% of patients. How was T1 stage defined?
Author Response
Dear reviewer:
We appreciate you much for your positive and constructive comments on our manuscript . We have studied the comments carefully and have made the revision.
- Comment: There are too many variables in significant in the nomogram.
Response: We agree with your viewpoint. But the D variable is a new concept that we put forward to express tumor volume. It was a try. It showed strong statistical significance in comparison of PSM and NSM and the univariate analysis. And it was retained in variable screening. So it's reasonable to believe that the D variable might have the potential to represent tumor volume. So we kept it. Of course, in the final model, it is not significant, which means that its value is not determined yet. Therefore, we hope that other researchers can verify the significance of the D variable. If it can be verified that the D variable is indeed a substitute for tumor volume, we believe it will greatly simplify some processes of the measurement and calculation. As for the NT variable, it has been eliminated in the univariate analysis. In terms of clinical application, too many variables do make the nomogram inconvenient. So we also provide a web calculator, hoping it can cover the shortage in this area.
2. Comment: How was T1 stage defined?
Response: cT-MRI group 1 included all patients with T2a or less. Or to say that most patients are in stage T2a. Since the number of T1 patients was small, separate grouping was not conducive to statistical analysis and was meaningless. So we combined them. Maybe our abbreviations are misleading. We will change the abbreviation from cT-MRI to T-MRI and illustrate the grouping definition again in Table 1.
We would like to express our great appreciation to you again.
Thank you very much.
Best wishes.
Ying Hao